# Behavioral and Proteomics Studies on the Regulation of Response Speed in Mice by Whey Protein Hydrolysate Intervention

**DOI:** 10.3390/nu17152500

**Published:** 2025-07-30

**Authors:** Xinxin Ren, Chao Wu, Hui Hong, Yongkang Luo, Yuqing Tan

**Affiliations:** 1Beijing Laboratory for Food Quality and Safety, College of Food Science and Nutritional Engineering, China Agricultural University, Beijing 100083, China; renxx4648@cau.edu.cn (X.R.); hhong@cau.edu.cn (H.H.); luoyongkang@cau.edu.cn (Y.L.); 2Research and Development, Hilmar Cheese Company, Hilmar, CA 95324, USA; cwu@hilmaringredients.com

**Keywords:** whey protein hydrolysates, reaction time, behavior, hippocampus, proteomics

## Abstract

**Background:** Response speed refers to an individual’s ability to perceive and react to harmful stimuli, which can vary due to genetics, neural regulation, and environmental factors. Our previous study demonstrated that whey protein hydrolysate was a potential means to enhance cognitive function. **Methods:** This study used a variety of behavioral methods to evaluate the functional effects of whey protein hydrolysate on improving reaction speed, and revealed its potential mechanisms through proteomics analysis. **Results:** The results showed that whey protein hydrolysate improved response speed in mice when tested against thermal pain, mechanical strength stimuli, and prepulse inhibition. Proteomic analysis of the hippocampus revealed changes in proteins related to arginine and proline metabolism, as well as neuroactive ligand–receptor interactions. **Conclusions:** These findings provide new insights into the neuromodulatory effects of whey protein hydrolysate and support its potential role in enhancing response speed and cognitive performance.

## 1. Introduction

Whey protein is a high-quality animal protein derived from the whey portion of milk. Its primary constituents include β-lactoglobulin, α-lactalbumin, glycopeptides, immunoglobulins, lactoferrin, and other bioactive components [1,2]. Whey protein is rich in branch-chain amino acids, which are crucial for muscle development, thus qualifying it as a high-quality protein source. Whey protein hydrolysate (WPH) is produced using enzymes and other biotechnological methods, and compared to native whey protein, this hydrolysate exhibits higher bioactivity, which has led to their growing attention in the fields of nutrition, sports science, and health. The distinct bioactive properties of whey protein hydrolysate demonstrate a broad spectrum of potential health benefits, particularly in enhancing immunity [3], antioxidation [4], anticancer effects [5], antidiabetic activity [6], and antihypertensive properties [7], as well as promoting muscle growth and maintaining gut health.

Speed quality is defined as the capacity of the human body to execute a specific action rapidly or traverse a defined distance within a specified time frame. It comprises reaction speed, movement speed, and the rate of displacement during cyclical movements. Among these, reaction speed is quantified by reaction time, defined as the duration between stimulus perception and motor response initiation. Reaction time serves as a physiological indicator for assessing neuromuscular excitability and overall bodily responsiveness. Research conducted at the University of London demonstrated that reaction speed is positively correlated with life expectancy and is an indicator of central nervous system function and cognitive agility. Individuals with the slowest reaction times face a twofold increase in premature mortality risk relative to those with average reaction speeds [8]. Slowed reaction speed is associated with neurological disorders, including Alzheimer’s and Parkinson’s diseases [9]. The regulation of reaction speed involves a multitude of signaling pathways and neurotransmitter systems, particularly the norepinephrine and N-methyl-D-aspartic acid (NMDA) receptor systems, which play integral roles in neurophysiological processes. Norepinephrine, a key neurotransmitter released by postganglionic sympathetic fibers, modulates alertness, attention, and cognition [10]. NMDA receptors are key receptors for synaptic transmission in the brain, especially in the cerebral cortex, hippocampus, and basal ganglia. Studies have shown that activation of NMDA receptors can increase the efficiency of synaptic transmission and promote the rapid response of neural circuits, thereby improving the body’s reaction speed [11].

Peptides in whey protein hydrolysate have demonstrated potential in various fields, particularly in neuroprotection, antioxidation, anti-inflammation activity, and enhancing neural conduction velocity, with notable attention given to their effects in alleviating cognitive impairments. Previous studies have shown that WPH, particularly rich in glycine-threonine-tryptophan-tyrosine (GTWY) peptides—a tetrapeptide derived from β-lactoglobulin during whey protein digestion—can promote neuroplasticity and cognitive performance improvements, including learning, memory consolidation, and executive function in older adults [12,13]. Furthermore, whey protein hydrolysate exhibits neuroprotective properties by modulating neurotransmitter release and attenuating neuroinflammatory responses [14]. While numerous studies have investigated the beneficial effects of whey protein hydrolysate on cognitive function, its influence on reaction speed regulation remains underexplored. Reaction speed, as a crucial functional indicator of the nervous system, is widely utilized to evaluate neuronal transmission efficiency and systemic health, particularly in the context of rapid responses to acute stimuli. Training interventions can improve reaction speed by enhancing neural plasticity, increasing synaptic efficiency, and promoting neuromuscular coordination. In particular, regular physical activity can optimize the function of the central nervous system, improving sensory-motor processing and facilitating faster decision-making and response times [15,16]. We hypothesize that whey protein hydrolysate (WPH) supplementation, when paired with training interventions, can significantly enhance reaction speed. A nociceptive-related behavioral assay was conducted, complemented by hippocampal proteomic analysis, to determine whether whey protein hydrolysate enhances reaction speed via improved neural conduction velocity or heightened neural excitability. These findings advance our understanding of whey protein hydrolysate’s potential applications in nervous system therapeutics.

## 2. Materials and Methods

### 2.1. Chemicals and Reagents

We used whey protein hydrolysate (Protelyze Bio-Brain, Hilmar, CA, USA), whey protein concentrate enriched in a milk fat globule membrane (Hilmar 7500, USA), phosphate-buffered saline (PBS; Sigma, Cream Ridge, NJ, USA), triethylammonium bicarbonate buffer (TEAB; Sigma, USA), Bond-Breaker™ TCEP Solution (Sigma, USA), iodoacetamide (IAM, Sigma, USA), modified trypsin (Promega, Madison, WI, USA), and the Pierce™ BCA Protein Assay Kit (Thermo Fisher Scientific, Waltham, MA, USA). “Whey Protein hydrolysate” and “whey protein concentrate enriched in milk fat globule membrane” are abbreviated as WPH and MFGM, respectively.

### 2.2. Animal Models

CD-1 mice (male, 6 weeks old, weighing 20 ± 2 g) were procured under specific pathogen-free (SPF) conditions from Vital River Laboratory Animal Technology Co., Ltd. (Beijing, China). All animal experiments were conducted in accordance with the Guide for the Care and Use of Laboratory Animals (8th ed., National Institutes of Health, USA) and approved by the Animal Ethics Committee of China Agricultural University (Approval No. AW41114202-4-3). Animal experiments strictly follow the 3R principle (Replacement, Reduction, and Refinement), minimizing the number of test animals and minimizing or eliminating pain without interfering with the research objectives. Mice were housed in the Chinese Institute for Brain Research, Beijing, under controlled conditions: constant temperature (22 ± 2 °C), a 12 h light–dark cycle, and 50% relative humidity.

Following a one-week acclimatization period, mice were randomly assigned to six groups (*n* = 6 per group): training and non-training groups, including a control group (phosphate-buffered saline, PBS), a whey protein hydrolysate (WPH) group, and a milk fat globule membrane (MFGM) group. The control group received daily oral gavage of PBS, while the WPH and MFGM groups were administered 2000 mg/kg body weight (BW) of their respective treatments once daily for six consecutive weeks. The training group underwent behavioral testing during weeks 1, 2, 4, and 6, whereas the non-training group was evaluated exclusively in week 6 (Figure 1A).

At the end of the experiment, the mice were euthanized, and serum, brain, hippocampus, liver, muscle, and selected tissues were aseptically collected. The muscle tissue was fixed in 4% paraformaldehyde, whereas the remaining tissues were snap-frozen in liquid nitrogen and subsequently stored at −80 °C.

### 2.3. Behavioral Tests

The mice in the training group underwent behavioral stimuli testing at 1, 2, 4, and 6 weeks to evaluate the potential positive effects of the training on the intervention’s efficacy. The non-training group was only tested in the sixth week. During the behavioral testing, a respective group of mice received oral gavages of PBS, WPH, or MFGM two hours before testing or training sessions. Before testing, mice were acclimated to the behavioral testing environment for 30 min. All mice underwent the three behavioral tests outlined in the three subsections below.

#### 2.3.1. Hot Plate Test

The mice were placed on a pre-heated thermal plate (BIO-HP, BIOSEB, Pinellas Park, FL, USA), maintained at 50 °C, a temperature sufficient to reliably induce nociceptive behavior. The latency of the pain response was recorded, defined as the time interval from placement on the dish to the first observation of paw-licking behavior. To minimize tissue damage, a 30 s cut-off time was strictly enforced, with animals failing to exhibit the target behavior within this period assigned the maximum latency value of 30 s [17]. The latency to pain response in different experimental groups was recorded as an indicator for assessing hot pain sensitivity.

#### 2.3.2. Von Frey Pain Experiment

The Von Frey test employs filaments of calibrated stiffness (typically nylon filaments) to deliver graded mechanical pressure, thereby evaluating an animal’s nociceptive response (such as hind paw withdrawal or licking). Mice were positioned on an elevated wire-mesh platform (BIO-EVF-WRS, BIOSEB, USA) and acclimatized for 15 min. Von Frey fibers were applied vertically from below the mesh to the plantar surface of the mouse’s hind paw, and the reaction threshold (defined as the minimum pressure that causes a response) of each mouse at different intensities was recorded to assess pain sensitivity. At the pain threshold, the mouse responds by flicking its paw away from the stimulus. Different pressure intensities (from light to heavy) are applied, with each fiber making contact for 2–3 s, observing the mouse’s pain response until the hind paw is withdrawn. During the formal experiment, a fixed-intensity Von Frey filament (0.1 g) was used to evaluate the withdrawal threshold of each mouse. The testing steps were repeated until all mice were tested, after which the procedure returned to the first mouse and was repeated. Measurements continued until the difference in intensity between two consecutive measurements was within 10%, at which point the data for each mouse were considered valid. The measurement was repeated three times for each mouse, with three 3 min intervals between repetitions. The apparatus platform was cleaned with alcohol after testing each animal to avoid odor interference.

#### 2.3.3. Prepulse Inhibition

The prepulse inhibition (PPI) experiment followed established protocols [18], with procedural optimizations implemented for this study. The startle system comprised four sound-attenuated chambers. Mice were placed in transparent cylindrical compartments, each equipped with a motion-sensitive sensor positioned beneath the chamber to quantify startle responses. Upon stimulation, movements generated pressure changes detected by the sensor, with signals digitized for computational analysis. Before the experiment began, the mice were placed in the testing environment for a 30 min acclimation period, allowing them to gradually adapt to the surroundings and minimize the influence of stress on the experimental results. The PPI experiment was conducted over two days to ensure that the mice could acclimate to various stimuli and establish baseline responses on the first day. On the second day, a fixed stimulus intensity was used to enhance the standardization and reproducibility of the experiment.

On Day 1, following a 5 min acclimation period under 65 dB background noise, startle stimuli (20 ms white noise) were delivered at random intervals (10–30 s). Stimulus intensity began at 70 dB and increased incrementally by 3 dB per trial, culminating at 130 dB across 20 trials. Data were analyzed to assess the differences in response times at various stimulus volumes, identifying the startle stimulus intensity that elicited the fastest response, thus providing insight into the mice’s sensitivity to various sound intensities. On Day 2, based on the adaptation response and baseline data from the first day, an appropriate fixed decibel level (106 dB, the sound intensity that elicited the shortest response time or the fastest startle response) was selected for standardized testing. After a 5 min acclimation period with a constant background white noise of 65 dB, a startle stimulus (20 ms white noise) was presented randomly every 10–30 s. A total of 10 startle stimulus trials were conducted. The data were analyzed to assess the differences between groups under this decibel level of sound stimulation. The data window was used to export the target data, and the average of all peak values for the fixed pulse on the second day was calculated as the PPI result. To minimize errors caused by physiological factors in the mice, the experiment was conducted between 9 a.m. and 2 p.m. To avoid odor interference, the testing chamber was cleaned with alcohol after each animal, and the testing room was kept quiet.

### 2.4. Hematoxylin–Eosin Staining

The gastrocnemius muscle, fixed in 4% paraformaldehyde, was dehydrated through an ethanol gradient, embedded in paraffin, and sectioned. The sections were then stained with hematoxylin–eosin (HE) staining to observe the structural changes in the mouse muscle fibers. Images were observed using a biological microscope (Nikon E400, Tokyo, Japan).

### 2.5. Hippocampal Proteomics Analyzing

#### 2.5.1. Total Protein Extraction

The frozen hippocampal tissue was transferred to a grinding tube, and we added an appropriate amount of protein lysis buffer (8 mol/L urea, 1% SDS, and protease inhibitors). The tissue was ground using a freezing grinder (90 Hz, 180 s), then incubated at 4 °C for 30 min. Afterward, it was centrifuged at 4 °C, 14,000× *g* for 15 min (Centrifuge 5430R, Eppendorf, Saxony, Germany) and we collected the supernatant.

#### 2.5.2. Protease Hydrolysis

We took 100 µg of the sample to be tested and added a solution of 100 mM TEAB and 10 mM TCEP to achieve the final concentrations. We mixed it thoroughly and incubated it at 37 °C for 60 min. Then, we added 40 mM IAM to the mixture and reacted it in the dark at room temperature for 40 min. After the reaction, it was centrifuged at 1000× *g* for 20 min at room temperature. We then collected the pellet and dissolved it in 100 µL of 100 mM TEAB. Then, we added trypsin at a 1:50 ratio and incubated it at 37 °C overnight for digestion.

#### 2.5.3. DIA Mass Spectrometry Detection

The enzyme-digested samples were concentrated, reconstituted, desalted, dried, and then reconstituted again. The final reconstituted samples were separated using the uPAC High-Throughput column (75 μm × 5.5 cm, Thermo, Waltham, MA, USA). The separation was performed with the VanquishNeo chromatograph (Thermo, USA) and the Astral mass spectrometer (Thermo, USA) in combination for LC-MS/MS analysis, with data collection and analysis conducted using Thermo Xcalibur 4.7.

### 2.6. Data Analysis

All data were calculated using GraphPad 8 (GraphPad Software Inc., La Jolla, CA, USA) software and analyzed using one-way ANOVA (Tukey’s multiple comparisons), and the results were expressed as the mean ± SD. Statistical differences were indicated when *p* < 0.05.

## 3. Results

### 3.1. The Effect of Whey Protein Hydrolysate (WPH) on Body Weight and Organ Index in Mice

During the experiment, mice were weighed weekly, and the data were plotted for analysis. As shown in Figure 1B, the weight of mice in each group showed an upward trend from 1 to 6 weeks, with no significant differences observed among the intervention groups. To further assess weight changes, the difference between the weight of mice in each group at 6 weeks and the initial weight were analyzed (Figure 1C), The results showed that there was no significant difference between the final body weight and the initial body weight, indicating that the intervention of MFGM and WPH did not induce obesity or other adverse effects. After 6 weeks of intervention with 2000 mg/kg BW WPH and MFGM, the organ coefficients (heart, liver, spleen, and kidney) of mice showed no significant changes compared with the control group, indicating that this dosage did not result in side effects or toxicity (Figure 1D–G). Compared with the control group, the brain coefficient of mice in the WPH intervention group was significantly higher (*p* < 0.05), suggesting that WPH supplementation may influence brain tissue function.

### 3.2. The Effect of WPH Intervention on the Reaction Speed of Mice

Three behavioral tests—the hot-plate test, the Von Frey mechanical pain test, and the prepulse inhibition (PPI) test—were used to evaluate the effects of WPH intervention and training on the reaction speed of mice (Figure 2). Compared with the PBS group, the WPH intervention shortened the reaction time of mice to heat pain in the hot-plate test by 14.6% (*p* < 0.05), and the WPH intervention and training treatment shortened the reaction time of mice to heat pain stimulation by 16.7% (*p* < 0.05), while there was no significant difference between the MFGM intervention group and the PBS group (Figure 2A). When mechanical pain stimulation was applied at an intensity of 0.1 g, only the mice in the intervention group combined with WPH and training showed significant differences. Specifically, the synergistic intervention of WPH and training significantly improved the response speed of mice to mechanical pain (*p* < 0.05), while the use of WPH alone did not show the same effect (Figure 2B). This suggests that the combined intervention of WPH and training may improve the response performance of mice under mechanical stimulation by increasing nerve conduction velocity. Moreover, in the prepulse inhibition experiment, the reaction time of mice was tested by 106 dB sound stimulation to evaluate their sensitivity to sound stimulation. The results showed that mice treated with WPH combined with training showed a significant shortening of reaction time when stimulated by sound waves at this decibel (Figure 2C, *p* < 0.05). In summary, WPH intervention and training can not only improve the response speed of mice to painful stimulation but also enhance their sensitivity to sound stimulation under non-painful stimulation. This finding further supports the positive role of the WPH in improving the speed of neural responses.

### 3.3. The Effect of WPH Intervention on Muscle Fiber Structure

Muscles play an important role in controlling the speed and strength of human movement. As a key skeletal muscle, the gastrocnemius is not only involved in movement coordination but also plays a core role in the sensory conduction, pain transmission, and pain maintenance of persistent nerve pain [19]. The HE staining results of the gastrocnemius muscle showed that, compared with the PBS group, the muscle fibers in the WPH and MFGM intervention groups were more closely arranged, the muscle striations were clear and regular, the number of myofiber nuclei increased, and the intercellular spaces decreased. These changes suggest that the WPH intervention may have improved the muscle tissue structure and enhanced the functional basis of the muscles. The training group can promote the orderly arrangement of muscle fibers to a certain extent. Although these structural improvements may have a positive effect on overall athletic performance, their relationship with the speed of nerve response still needs further research to clarify (Figure 3).

### 3.4. The Effect of WPH Intervention on Protein Expression in the Hippocampus of Mice

The hippocampus, as one of the important structures of the brain, not only participates in learning, memory, spatial navigation, and emotional regulation, but also has strong neural plasticity, enabling it to quickly adjust the connectivity of its neural networks in response to external environmental stimuli [20]. The hippocampus indirectly responds to pain stimuli by regulating the activity of the prefrontal cortex and basal ganglia [21]. When the neural excitability of the hippocampus increases, it helps enhance the efficiency of cortical processing of pain signals, thereby accelerating the reaction speed.

In this study, a DIA-based quantitative approach was used to investigate the changes in the hippocampal proteome of mice following WPH intervention. In the volcano plot analysis of differentially expressed proteins, proteins with a *p*-value < 0.05 and a fold change (FC) > 1.5 were considered significantly altered. In the MFGM intervention group, 103 proteins were upregulated and 132 were downregulated; in the WPH intervention group, 87 proteins were upregulated and 130 were downregulated. A total of 107 proteins showed similar regulatory trends in both intervention groups. To gain a comprehensive understanding of the biological functions of these differentially expressed proteins, GO enrichment analysis was performed on both intervention groups compared with the control. This analysis identified biological processes (BPs), cellular components (CCs), and molecular functions (MFs). Figure 4C shows the top 10 terms for the MFGM group. In the BP category, the terms were primarily related to metabolic processes. The CC category analysis indicated that most proteins were predicted to be localized in the nucleus and chromosomes. The MF analysis revealed that most proteins were associated with the binding of glucose, metal ions, organic acids, and enzyme catalytic activity. In the WPH intervention group, the BP category showed involvement in the metabolic processes of glucose and organic acids. In the CC category, the proteins were mainly localized to the nucleus and the cell membrane. The MF analysis indicated that these proteins were primarily involved in enzyme catalytic activity, DNA binding, and immunoglobulin receptor binding.

KEGG pathway enrichment showed (Figure 5A,B) that in the MFGM intervention group, 15 of the 24 significantly enriched pathways were associated with metabolic processes. The differentially expressed proteins were linked to amino acid metabolism, including tyrosine metabolism and tryptophan metabolism, as well as the pentose phosphate pathway. In the WPH intervention group, 16 significantly enriched pathways were identified, with 11 of them involved in metabolic processes. The differential proteins were related to arginine and proline metabolism, as well as the biosynthesis of mannose-type O-glycosylation. The chord diagram (Figure 5C,D) shows the relationships between different KEGG pathways and their connections with specific genes. After MFGM treatment, key genes such as Adh1, Pah, and Aox3 were found to be closely associated with amino acid metabolism pathways. In contrast, following the WPH intervention, key genes such as Arg1 and L3hypdh were predominantly involved in arginine and proline metabolism.

We further analyzed the impact of training on differentially expressed proteins in the hippocampus, comparing the enriched pathways between trained and non-trained groups under PBS, MFGM, and WPH interventions (Figure 5E). In the PBS group, training led to significant enrichment in amino acid metabolism pathways, including tryptophan and tyrosine metabolism, suggesting that training may enhance these pathways, thereby improving neurotransmitter synthesis and neural signal transmission. In the MFGM and WPH intervention groups, training primarily enriched pathways related to arginine biosynthesis, steroid hormone biosynthesis, and glycosylation modifications (O-glycosylation), indicating that training might modulate neural function and cellular repair through these metabolic pathways. Notably, both retinol metabolism and neuroactive ligand–receptor interaction pathways were significantly enriched across all treatments, suggesting that these pathways may be crucial for training-induced improvements in reaction speed and cognitive ability in mice.

## 4. Discussion

In this study, we investigated the modulatory effects of whey protein hydrolysate (WPH), enriched with the bioactive tetrapeptide GTWY (β-lactoprotein), on reaction speed in mice. Our results suggest that WPH, especially in combination with training, significantly enhances response speed to various stimuli, including thermal, mechanical, and auditory cues. This improvement may be linked to enhanced neuroplasticity, altered neurotransmitter metabolism, and structural changes in brain and muscle tissues.

The brain is the most metabolically active organ, and an instantaneous response speed involves a complex network of brain regions, including the thalamus, insula, somatosensory cortex, anterior cingulate cortex, and prefrontal cortex [22]. Our observation that the WPH intervention increased the brain coefficient in mice implies potential changes in brain structure or function. Previous studies have reported that WPH improves cognitive function and memory in aging mice, which aligns with our hypothesis that WPH may also contribute to accelerated neural responses [12].

Reaction speed is a key behavioral parameter that reflects an organism’s capacity to process external stimuli and execute rapid motor responses. In this study, we assessed motor response latency using the hot-plate test and Von Frey stimulation and evaluated auditory sensorimotor gating with prepulse inhibition (PPI), providing a comprehensive understanding of the neural mechanisms underlying reaction speed. PPI is a well-established neurobehavioral paradigm that assesses the brain’s ability to filter incoming information and regulate sensorimotor gating. Previous studies have shown that PPI is modulated by subcortical circuits, as well as higher-order cortical regions, including the prefrontal cortex, hippocampus, and amygdala [23]. These brain regions, through neurotransmitters like glutamate and γ-aminobutyric acid, coordinate the timing and accuracy of stimulus processing, forming the neurobiological foundation of reaction speed [24]. Furthermore, neurobehavioral training has been demonstrated to enhance reaction speed. Our study found that WPH intervention and training significantly improved the reaction speed of mice under various stimuli, likely by enhancing neurotransmitter regulation, promoting synaptic plasticity, and improving the efficiency of central information integration. Consistent with findings in human studies, athletes have shown a significant reduction in reaction time through reaction time training, indicating that physical training can enhance the central nervous system’s processing efficiency and motor readiness in response to external stimuli [25]. This improvement in reaction speed may not solely be attributed to short-term neural reflex training but is likely associated with long-term neuroadaptive changes, such as structural and functional alterations in synaptic connectivity.

In the context of whey protein hydrolysates (WPHs), peptides, as key hydrolysis products, have demonstrated notable neuroprotective effects in initial studies. Specific peptides, such as GTWY, present in the WPH have shown potential in facilitating neurotransmitter conduction [12]. Oral administration of GTWY significantly increased hippocampal extracellular dopamine concentrations, thereby improving spatial working memory and reference memory. This effect was found to be dependent on the activation of D1 receptors [26]. These peptides may interact with neuronal receptors or influence signaling pathways to enhance neurotransmission efficiency. Although the exact mechanisms underlying the action of GTWY and its impact on amino acid metabolism remain to be fully elucidated, the preliminary evidence suggests its promising role in supporting brain health and function.

In KEGG pathway enrichment analysis, we observed significant enrichment in the neuroactive ligand–receptor interaction pathway in both training and non-training groups, further supporting this hypothesis. This pathway involves various neurotransmitters, such as dopamine, serotonin (5-HT), and glutamate, which activate downstream signaling pathways through their specific receptors (the D1 receptor, 5-HT receptor, and NMDA receptor). This activation enhances synaptic transmission efficiency and the strength of neural signal conduction [27]. Furthermore, exercise is known to enhance neuroplasticity, increasing neurogenesis, cerebral blood flow, and neurotransmitter release, including dopamine and serotonin [28,29]. Exercise elevates the levels of catecholamine neurotransmitters, which enhances their release and strengthens neural signal transmission, further promoting hippocampal synaptic plasticity and neurogenesis [30]. This mechanism explains why physical training can effectively enhance animals’ reaction speed and cognitive abilities (Figure 6). Our results emphasize that combining WPH, particularly enriched GTWY peptides, with training amplifies neurobiological mechanisms related to amino acid metabolism, neurotransmitter synthesis, and neuroplasticity pathways, thus significantly improving reaction speed and sensitivity. Specifically, WPH enhances neurotransmitter synthesis and transmission, elevating neuronal responsiveness in the hippocampus, thereby facilitating rapid neural responses during training interventions.

## 5. Conclusions

In summary, the intervention of WPH enriched in GTWY combined with training can increase the response speed of mice to pain stimuli. This may be related to its regulation of arginine metabolism, release of nitric oxide, and positive modulation of dopamine and serotonin to promote synaptic plasticity in neurons. However, further investigation into the neurochemistry associated with the neurotransmitter system is necessary to clarify the potential mechanisms through which WPH and training regulate response speed.

## Figures and Tables

**Figure 1 nutrients-17-02500-f001:**
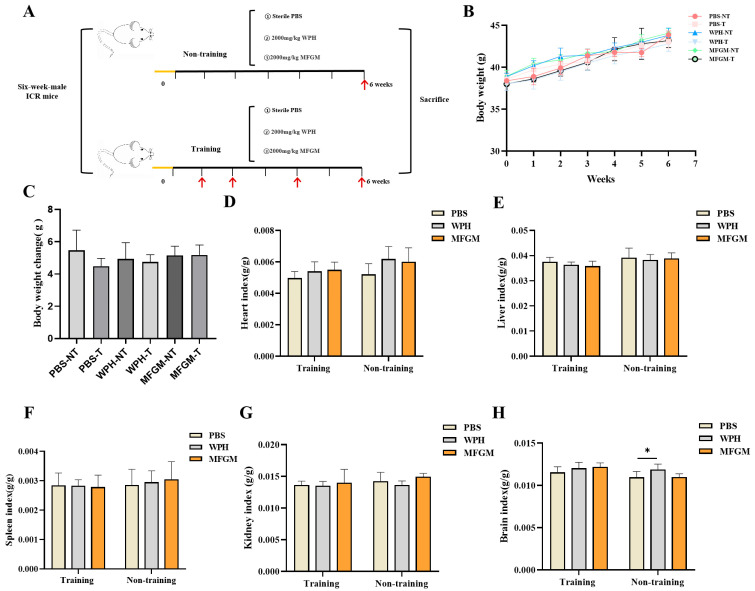
Effects of different interventions on the body weight and organ index of mice. Experiment flowchart, the red arrows indicate the time for behavioral training (**A**), body weight (**B**), difference between initial weight and final weight (**C**), organ coefficient: heart (**D**), liver (**E**), spleen (**F**), kidney (**G**), brain (**H**). Results are shown as mean ± SD. * *p* < 0.05, PBS vs. WPH, PBS vs. MFGM.

**Figure 2 nutrients-17-02500-f002:**
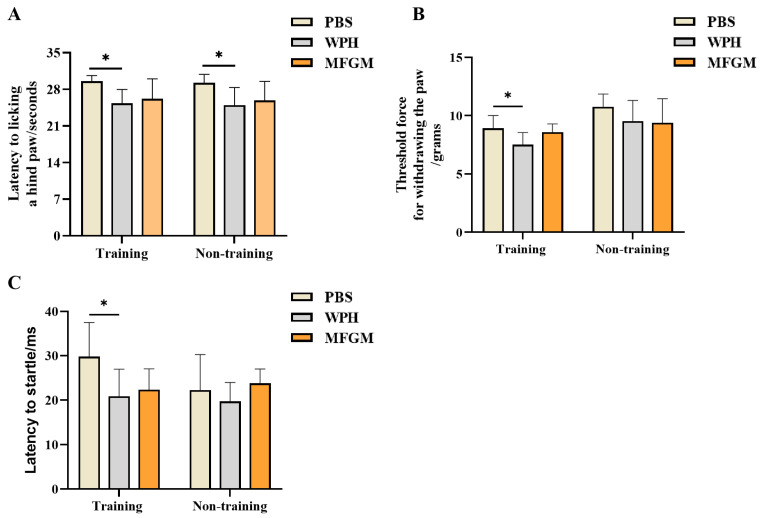
Effects of whey protein hydrolysate intervention on the reaction speed of mice. Hot-plate experiment (**A**), Von Frey Pain Experiment (**B**). Prepulse inhibition (**C**). Results are shown as mean ± SD. * *p* < 0.05, PBS vs. WPH, PBS vs. MFGM.

**Figure 3 nutrients-17-02500-f003:**
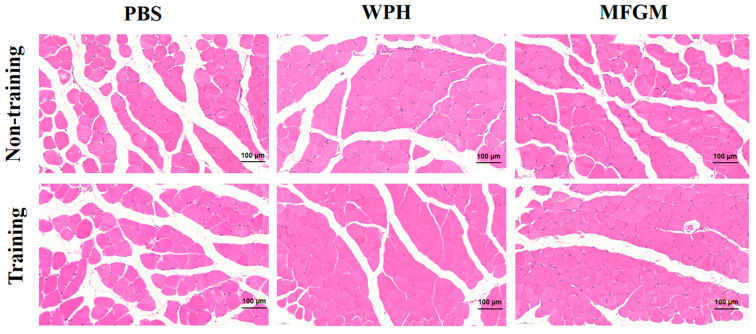
H&E staining was used to observe the morphological changes in the muscles of mice in different treatment groups.

**Figure 4 nutrients-17-02500-f004:**
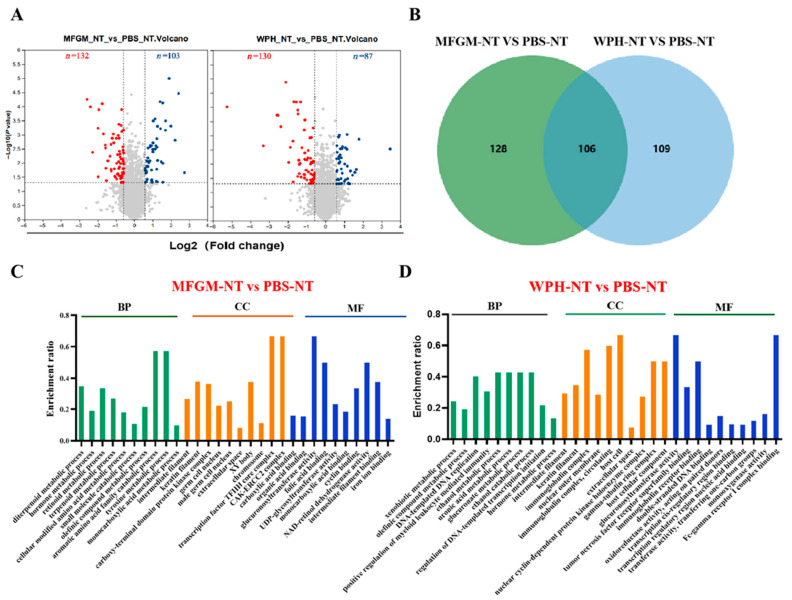
Proteomic analysis. Protein volcano scatter plot (**A**). Venn diagram (**B**). GO enrichment analysis (**C**,**D**).

**Figure 5 nutrients-17-02500-f005:**
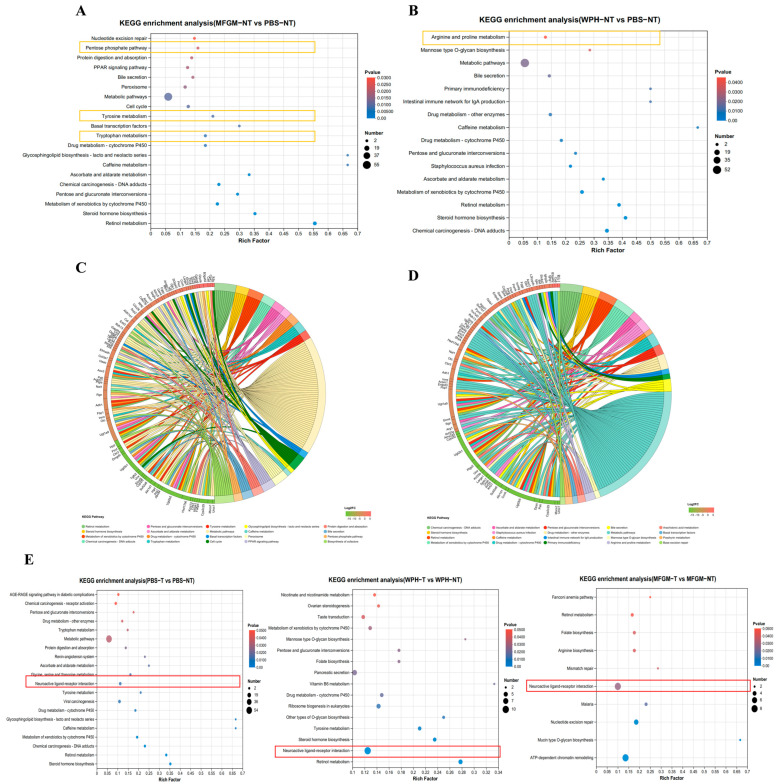
KEGG pathway enrichment in proteomics. The bubble plots (**A**,**B**,**E**) and chord diagrams (**C**,**D**) of differential proteins in KEGG signaling pathways.

**Figure 6 nutrients-17-02500-f006:**
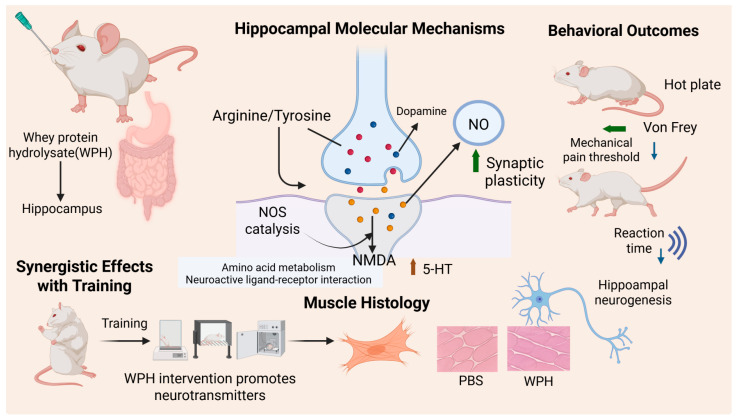
Whey protein hydrolysate intervention promotes neurotransmitter release and amino acid metabolism, enhances hippocampal synaptic plasticity and neurogenesis, and synergizes with training to improve central nervous system function and behavioral responses.

## Data Availability

The data presented in this study are available on request from the corresponding author. The data are not publicly available due to the laboratory confidentiality policy.

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
