# Peer review of "Behavioral and Proteomics Studies on the Regulation of Response Speed in Mice by Whey Protein Hydrolysate Intervention"

_nutrients, 2025, doi:10.3390/nu17152500_

Round 1
Reviewer 1 Report
Comments and Suggestions for Authors
Dear Authors,
The manuscript presents a well-structured and methodologically sound investigation into the potential effects of whey protein hydrolysate (WPH) on reaction speed in mice, supported by behavioral tests and proteomic analysis. The topic is timely and relevant, particularly in the context of growing interest in functional foods and nutraceuticals. The results are of potential interest to readers in the fields of nutrition, neuroscience, and functional food development.
However, to improve the manuscript, some clarifications and corrections are necessary:
-
Clarity of Hypothesis and Aim
-
The hypothesis is implied but not clearly articulated. The authors should more explicitly state their central hypothesis in the introduction (e.g., "We hypothesize that WPH improves response speed by modulating neuroplasticity and neurotransmitter pathways").
-
-
Terminology Consistency
-
There is occasional inconsistency in referring to the experimental groups (e.g., "training group," "synergistic training," "trained group"). Consider standardizing terms throughout.
-
-
Statistical Analysis
-
While the use of one-way ANOVA is appropriate, the manuscript lacks detailed reporting of statistical values (e.g., F-values, degrees of freedom). These should be included either in the main text or supplementary tables.
-
It would be useful to report effect sizes where appropriate.
-
-
Proteomics Interpretation
-
The link between differentially expressed proteins and improvements in reaction speed is intriguing but remains speculative. A stronger discussion of causality vs. correlation is warranted.
-
Some pathways are discussed generally (e.g., “enzyme catalytic activity” or “glucose metabolism”) without direct relevance to neural function. Focus more sharply on neurorelevant mechanisms.
-
-
English Language
-
Some sentences are long and awkwardly constructed.
-
A thorough proofreading by a native English speaker or editing service is recommended.
-
6. Muscle fiber analysis: While the HE staining results are interesting, it is unclear how changes in muscle fiber arrangement relate directly to neural response speed. This section would benefit from a more cautious interpretation.
7. Training effects: The distinction between the effects of WPH alone and WPH with training is crucial. However, the manuscript occasionally conflates the two. Ensure that comparisons are made clearly and only between relevant groups.
Kind regards
Comments on the Quality of English Language
English Language
-
-
Some sentences are long and awkwardly constructed.
-
A thorough proofreading by a native English speaker or editing service is recommended.
-
Author Response
Comments 1:Clarity of Hypothesis and Aim:The hypothesis is implied but not clearly articulated. The authors should more explicitly state their central hypothesis in the introduction (e.g., "We hypothesize that WPH improves response speed by modulating neuroplasticity and neurotransmitter pathways"). |
Response 1:Thank you for pointing this out. We agree with this comment. Therefore, we have added emphasis on the research purpose of this study in lines 73-79 of the introduction. [Training interventions can improve reaction speed by enhancing neural plasticity, increasing synaptic efficiency, and promoting neuromuscular coordination. In particular, regular physical activity can optimize the function of the central nervous system, improving sensory-motor processing and facilitating faster decision-making and response times[15, 16]. we hypothesize that whey protein hydrolysate (WPH) supplementation, when paired with training interventions, can significantly enhance reaction speed. ] |
Comments 2:Terminology Consistency:There is occasional inconsistency in referring to the experimental groups (e.g., "training group," "synergistic training," "trained group"). Consider standardizing terms throughout. |
Response 2: Thank you for pointing this out. We have unified the description. Modifications have been made in lines 246[intervention and training ], 260[intervention and training], 340[ non-training], and 401[training and non-training] . |
Comments 3:Statistical Analysis:While the use of one-way ANOVA is appropriate, the manuscript lacks detailed reporting of statistical values (e.g., F-values, degrees of freedom). These should be included either in the main text or supplementary tables. It would be useful to report effect sizes where appropriate. |
Response 3: Thank you for pointing this out. According to the reviewer's suggestion, we have added the F value and degrees of freedom of the groups with significant differences in the table below so that the reviewer can view the detailed data. |
Comments 4:Proteomics Interpretation:The link between differentially expressed proteins and improvements in reaction speed is intriguing but remains speculative. A stronger discussion of causality vs. correlation is warranted. Some pathways are discussed generally (e.g., “enzyme catalytic activity” or “glucose metabolism”) without direct relevance to neural function. Focus more sharply on neurorelevant mechanisms. |
Response 4: We sincerely appreciate your insightful comments regarding the need to clarify causality and to focus more specifically on neurologically relevant pathways. With respect to the enrichment of general terms such as "catalytic activity" and "glucose metabolism," these annotations primarily emerged from Gene Ontology (GO) molecular function analysis, which reflects the general biochemical properties of the differentially expressed proteins. However, as rightly pointed out, such broad terms may not directly illuminate the neural mechanisms underlying behavioral changes. Therefore, in the KEGG pathway enrichment analysis, we deliberately focused on pathways more closely related to neural function, including amino acid metabolism and neurotransmitter-related signaling cascades, to provide more biologically relevant insights into the observed enhancement in reaction speed. We also acknowledge that statistical associations alone cannot establish causality between protein expression changes and behavioral outcomes. In future work, we aim to incorporate functional validation experiments, such as neurobehavioral assays and targeted protein perturbation, to further elucidate the causal roles of these candidate proteins. Thank you again for your constructive suggestions, which help us further focus and deepen the research content. |
Comments 5:English Language:Some sentences are long and awkwardly constructed. A thorough proofreading by a native English speaker or editing service is recommended. |
Response 5: We are very grateful for your careful review and constructive suggestions on the language quality of the manuscript. In response, we have comprehensively revised the text, paying particular attention to sentence clarity, conciseness and overall readability, which have been marked in the manuscript with a yellow highlighter. Thank you again for pointing out this important issue, which has greatly helped us improve the clarity and presentation of the manuscript. lines 30、135-137、147-150、227-228、229-234、236-240、242-243、253、299-306、321-325、327、330-333、339-340、355、367-375、378-385、391-392、394-395、404-406、409、414. |
Comments 6:Muscle fiber analysis: While the HE staining results are interesting, it is unclear how changes in muscle fiber arrangement relate directly to neural response speed. This section would benefit from a more cautious interpretation. |
Response 6:Thank you very much for your valuable suggestions. We agree with your point that there is no direct relationship between changes in muscle fiber arrangement and response speed. We have modified the relevant statements to avoid over-interpreting the direct effects of tissue morphology changes on neural responses (lines 270-282). We emphasized in the article that HE staining results mainly reflect the improvement of muscle tissue structure, especially the compactness and orderliness of muscle fiber arrangement, which may indirectly support the improvement of athletic performance, but it is currently impossible to confirm that there is a direct causal relationship between it and the speed of nerve reaction. Thank you for your rigorous review and constructive feedback, which made our research expression clearer and more accurate. |
Comments 7:Training effects: The distinction between the effects of WPH alone and WPH with training is crucial. However, the manuscript occasionally conflates the two. Ensure that comparisons are made clearly and only between relevant groups. |
Response 7:Thank you for your valuable suggestions. We recognize the importance of distinguishing the effects of WPH alone and its combined effects with training. To ensure a clear comparison of these two conditions, we have made revisions in each results section to focus on comparisons between relevant groups to avoid confounding between the effects of WPH alone and its combined effects with training. We believe that these revisions can address your concerns and help clarify the effects of different interventions in our analysis. Thank you again for your valuable suggestions. |
4. Response to Comments on the Quality of English Language |
Point 1: Some sentences are long and awkwardly constructed. A thorough proofreading by a native English speaker or editing service is recommended. |
Response 1: We are very grateful for your careful review and constructive suggestions on the language quality of the manuscript. In response, we have comprehensively revised the text, paying particular attention to sentence clarity, conciseness and overall readability, which have been marked in the manuscript with a yellow highlighter. Thank you again for pointing out this important issue, which has greatly helped us improve the clarity and presentation of the manuscript. (in red) |
Reviewer 2 Report
Comments and Suggestions for Authors
This study aims to assess their functional effects on improving response speed by whey protein hydrolysates intervention and to uncover the underlying mechanisms through proteomic analysis. The results of this study showed that whey protein hydrolysate improved response speed in mice when tested against thermal pain, mechanical strength stimuli, and prepulse inhibition. In addition, Proteomic analysis of the hippocampus revealed changes in proteins related to arginine and proline metabolism, as well as neuroactive ligand-receptor interactions. The reviewer believes that the results of this study indicate that whey protein hydrolysates are effective in improving the response speed, which may contribute to the potential applications of whey protein hydrolysates in the future. However, there are several limitations in this study.
- In the “Methods” section, the authors describe ethical committee approval. However, the authors did not explain the animal welfare considerations in conducting the animal experiments. This article does not describe whether the animal experiments were conducted in accordance with the 3R principles: "1. Replacement of animals with alternatives wherever possible", "2. Reduction in the number of animals used", and "3. Refinement of experimental conditions and procedures to minimize the harm to animals". The authors should include a description of animal welfare considerations in conducting animal studies.
- This study aims to assess their functional effects on improving response speed by whey protein hydrolysates intervention. This study examined the combined effects of whey protein hydrolysate administration and training intervention. However, in the “Introduction” section, the authors did not explain the significance and purpose of training in addition to whey protein hydrolysate administration. In the “Introduction” section, authors should explain the significance and purpose of training in addition to whey protein hydrolysate administration.
- In relation to the above, in the “Methods” section, the authors do not provide any explanation for the content of the training. The authors should provide a description of the training provided in this study in the “Methods” section.
- In this study, the training group underwent behavioral testing during weeks 1, 2, 4, and 6, whereas the non-training group was evaluated exclusively in week 6. Why was behavioral testing only performed on the non-training group before and six weeks after the intervention in this study? The reviewer considers that if a comparison is to be made with a training group, the same protocol should be used.
- Could the results of this study be applied to humans in the future? Please explain authors’ future outlook for how the results of this study will be applied clinically.
Author Response
Comments 1: In the “Methods” section, the authors describe ethical committee approval. However, the authors did not explain the animal welfare considerations in conducting the animal experiments. This article does not describe whether the animal experiments were conducted in accordance with the 3R principles: "1. Replacement of animals with alternatives wherever possible", "2. Reduction in the number of animals used", and "3. Refinement of experimental conditions and procedures to minimize the harm to animals". The authors should include a description of animal welfare considerations in conducting animal studies. |
Response 1: Thank you for your attention and reminder about the animal ethics issues in this study. We take this very seriously. During the experiment, we strictly followed the "3R" principle of animal experiment ethics, namely "Replacement, Reduction and Refinement", and strived to minimize the number of experimental animals used and the suffering they endured while ensuring the scientific nature of the experiment. In addition, we have added information on animal welfare and ethical approval in lines 97-102 of the paper to further clarify our compliance with the ethical standards of animal experiments. Thank you again for your concern and cautious attitude towards animal ethics, which is of great significance to our improvement of research. |
Comments 2: This study aims to assess their functional effects on improving response speed by whey protein hydrolysates intervention. This study examined the combined effects of whey protein hydrolysate administration and training intervention. However, in the “Introduction” section, the authors did not explain the significance and purpose of training in addition to whey protein hydrolysate administration. In the “Introduction” section, authors should explain the significance and purpose of training in addition to whey protein hydrolysate administration. |
Response 2:Thank you for your careful review of the structure and content of the introduction. In response to the issues you pointed out, we have supplemented the background and research significance of the training intervention in the revised manuscript(lines 73-79). The important role of the training intervention in this study was clarified, aiming to simulate a physiological state closer to actual athletic performance and explore the synergistic or promoting effect of whey protein hydrolysate in functional improvement. Thank you again for your valuable suggestions, which helped us further improve the logic and content expression of the article. |
Comments 3:In relation to the above, in the “Methods” section, the authors do not provide any explanation for the content of the training. The authors should provide a description of the training provided in this study in the “Methods” section. |
Response 3:Thank you very much for your attention and valuable suggestions for this study. In response to the problem you pointed out that the "Methods" section did not describe the training content, we explained it in lines 106-113 of the revised manuscript, describing in detail the behavioral training arrangements for the mice in the training group, including the training cycle, training form, and specific implementation plans for each stage. Specifically, the mice received systematic behavioral training in weeks 1, 2, and 4 to help them gradually adapt to the experimental environment and establish relatively stable training responses, so as to more reliably evaluate the role of whey protein hydrolysate intervention in improving reaction speed. Thank you again for your constructive suggestions, which helped us further improve the scientificity and repeatability of the methods section. |
Comments 4:In this study, the training group underwent behavioral testing during weeks 1, 2, 4, and 6, whereas the non-training group was evaluated exclusively in week 6. Why was behavioral testing only performed on the non-training group before and six weeks after the intervention in this study? The reviewer considers that if a comparison is to be made with a training group, the same protocol should be used. |
Response 4: Thank you very much for your valuable comments on the experimental design of this study. We fully understand your concern about the issue that the non-training group only underwent behavioral testing at week 6, and would like to further explain the design ideas here. The original intention of this study was to evaluate the independent and combined effects of WPH intervention and behavioral training on reaction speed. The training group received behavioral tests at weeks 1, 2, 4, and 6. The main purpose was to dynamically monitor the changes in the reaction speed of mice during the training adaptation process in conjunction with the training process, so as to more accurately evaluate the training intervention itself and its interactive effects with WPH. In contrast, the non-training group did not undergo systematic training, so we arranged for them to undergo behavioral tests only before the intervention and at the end of the sixth week, aiming to observe whether WPH alone can produce significant endpoint improvement effects within 6 weeks without training intervention. We sincerely accept your suggestions on the consistency of test time between groups. In subsequent studies, we will further optimize the experimental process and consider making the test arrangements of each group more unified on the basis of controlling the potential training effect, so as to improve the comparability of data and the rigor of the study. Thank you again for your careful review and professional guidance of our work. |
Comments 5:Could the results of this study be applied to humans in the future? Please explain authors’ future outlook for how the results of this study will be applied clinically. |
Response 5: Thank you very much for raising this crucial and forward-looking question. This study aims to explore the effects of WPH combined with training intervention on the reaction speed of mice. Although the research subjects are animal models, the results obtained provide a valuable theoretical basis for potential applications in humans in the future. As a natural nutritional supplement rich in bioactive peptides, WPH has shown its positive effects in anti-oxidation, anti-inflammation and regulation of nerve conduction in many studies. The potential of WPH observed in this study in improving nerve reaction speed and promoting neuroplasticity suggests that it is expected to play a role in promoting human neurological function recovery, optimizing sports performance and even improving certain neurocognitive disorders in the future, especially in the aging or functional degeneration population. However, we are also well aware that there is still a certain gap between animal experiments and clinical applications. Therefore, we remain cautious about the extrapolation of the results of this study, and believe that systematic interventional clinical studies in humans are still needed in the future to verify its safety and effectiveness, so as to promote its translational application in actual health intervention or clinical adjuvant therapy. Thank you again for your attention and guidance on the clinical significance of this study. |
4. Response to Comments on the Quality of English Language |
Point 1: |
Response 1: (in red) |
Round 2
Reviewer 2 Report
Comments and Suggestions for Authors
I think all responses to reviewers' comments have been addressed satisfactorily.
I have no comments on the revised manuscript.